# TC-Bench: Benchmarking Temporal Compositionality in Conditional Video Generation

## Abstract

Video generation has many unique challenges beyond those of image generation. The temporal dimension introduces extensive possible variations across frames, over which consistency and continuity may be violated. In this study, we move beyond evaluating simple actions and argue that generated videos should incorporate the emergence of new concepts and their relation transitions like in real-world videos as time progresses. To assess the **T**emporal **C**ompositionality of video generation models, we propose **TC-Bench**, a benchmark of meticulously crafted text prompts, corresponding ground truth videos, and robust evaluation metrics. The prompts articulate the initial and final states of scenes, effectively reducing ambiguities for frame development and simplifying the assessment of transition completion. In addition, by collecting aligned real-world videos corresponding to the prompts, we expand TC-Bench's applicability from text-conditional models to image-conditional ones that can perform generative frame interpolation. We also develop new metrics to measure the completeness of component transitions in generated videos, which demonstrate significantly higher correlations with human judgments than existing metrics. Our comprehensive experimental results reveal that state-of-the-art video generators achieve less than 20% of the compositional changes, highlighting enormous space for improvement. Our analysis indicates that current video generation models struggle to interpret descriptions of compositional changes and synthesize various components across different time steps.

## 1 Introduction

Conditional video generation is the task of synthesizing realistic videos based on controlling inputs such as text prompts (text-to-video, T2V) or images (image-to-video, I2V). Significant advancement in dataset scale and model design has led to several large-scale, high-quality video generation models, such as CogVideo (Hong et al., 2022), VideoCrafter (Chen et al., 2023a), Stable Video Diffusion (Blattmann et al., 2023a), and others (Ho et al., 2022; Singer et al., 2022; Blattmann et al., 2023b). The additional time dimension in videos makes it essential to accurately assess and benchmark the alignment between the generated temporal variations and the condition inputs. While several studies have proposed fine-grained and comprehensive evaluation protocols (Huang et al., 2024c; Liu et al., 2024c;b; Wu et al., 2024), *compositionality in the temporal dimension* remains an under-addressed yet crucial aspect of video generation tasks.

The principle of compositionality specifies how constituents are arranged and combined to make a whole (Bienenstock et al., 1996; Partee, 2008; Cresswell, 2016). Ideal generative systems should produce outputs that reflect the compositions described by the prompts (Liu et al., 2022; Li et al., 2023a; Dziri et al., 2024). In image generation, prior work has focused on improving faithful compositionality in attributes, numbers, and spatial arrangement (Feng et al., 2022; Chatterjee et al., 2024; Lee et al., 2023). In video generation, compositional faithfulness is much more challenging— the output must consistently reflect the required combination of concepts, even as it changes through time. In this work, we investigate this *temporal compositionality* problem in video generation models by focusing on prompts describing scenarios where object attributes or relations change over time.

While image generation prompts involving spatial compositionality (Yu et al., 2022; Huang et al., 2024b) and video prompts describing actions or motions (Huang et al., 2024c; Liu et al., 2024c; Soomro et al., 2012; Xu et al., 2016) have been used for assessing T2V models, they have two

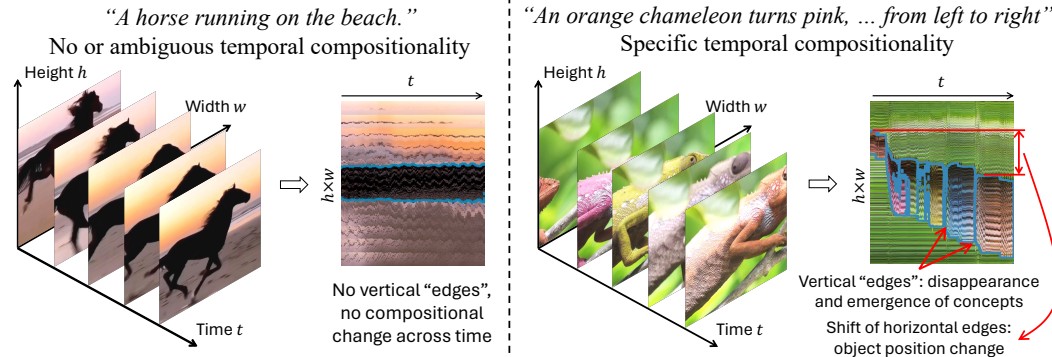

Figure 1: **Left**: a common text-video pair used in video generation evaluation with no temporal compositionality. **Right**: a sample from our TC-Bench. Different colors of the chameleon are composed along the time axis, resulting in the vertical "edges" in the spatiotemporal image. The gap between horizontal edges shows changes in the chameleon's position and its relation with the branch.

drawbacks: first, these prompts describe invariant compositions in time, and second, they lead to synthesized videos that manipulate existing metrics. For instance, while Fig. 1 (left) depicts "a horse running on the beach" motion, there are no compositional variations in the visual entities along the time axis. Such omissions can lead to flaws that, while noticeable to human users, are not captured by current benchmarks. In contrast, Fig. 1 (right) involves more specific compositional changes in position and color, marked by the vertical "edges" and the gap between horizontal edges in the spatiotemporal image representing attribute or object binding changes.

To this end, we propose **T**emporal **C**ompositionality **Bench**mark (**TC-Bench**), which addresses three scenarios of compositional changes: attribute transition, object relations, and background shifts. We craft realistic prompts that clearly specify an object's initial and final states, thereby requiring changing compositional characteristics in a correctly synthesized video. These prompts span a wide range of topics and scenes and present distinct challenges to different modules of T2V models. On the one hand, the text encoding stage needs to aggregate different groups of constituents from the prompt to guide the generation of different frames. On the other hand, the generation module must synthesize seamless transitions between frames while maintaining object consistency. To broaden applicability to I2V, we collect ground truth videos corresponding to the prompts, which allows us to benchmark models capable of performing generative frame interpolation (Chen et al., 2023b; Xing et al., 2023).

To facilitate the use of TC-Bench, we propose two evaluation metrics, TCR and TC-Score, that first produce frame-level compositionality assertions and check them throughout the video using vision language models (VLMs). TCR and TC-Score measure compositional transition completion and overall text-video alignment, which are better correlated to human judgments than existing metrics. We extensively benchmark multiple baselines across three categories of methods, ranging from direct T2V models (Wang et al., 2023a; Chen et al., 2024; Zhang et al., 2023; Wang et al., 2023b) to multi-stage T2V (Huang et al., 2024a; Lian et al., 2023) and I2V models (Chen et al., 2023b; Xing et al., 2023). Our comprehensive experiments demonstrate that most of the video generation models accomplish less than ∼20% of the test cases, implying enormous space for future improvement. Our contribution can be summarized as three points:

- TC-Bench, a new benchmark that characterizes temporal compositionality in video generation. TC-Bench features different types of realistic transitions and covers a wide range of visual entities, scenes, and styles.

- We propose new metrics to evaluate transition completion and text-video alignment and investigate consistency measures with various methods. Our metrics achieve much higher correlations with human judgments for evaluating temporal compositionality.

- A comprehensive evaluation of nine baselines shows that existing T2V and I2V methods still struggle with temporal compositionality. Our in-depth analysis reveals key weaknesses of current methods in prompt understanding and maintaining temporal consistency.

## 2 RELATED WORK

### 2.1 CONDITIONAL VIDEO GENERATION

Conditional video generation has been a challenging task (Balaji et al., 2019; Zhang et al., 2022; Fu et al., 2023; Blattmann et al., 2023a). Recently, with the advancement of diffusion models (Ho et al., 2020; 2022) and large-scale video datasets (Bain et al., 2021; Wang et al., 2023c), video generation models have gained significant improvement (Ho et al., 2022; Singer et al., 2022). Several studies attempt to add temporal operation layers into a pre-trained image model, such that the latter can be adopted as a video generation model in a zero-shot manner (Khachatryan et al., 2023) or through fine-tuning (Blattmann et al., 2023b). The idea of latent space diffusion (Rombach et al., 2021) has also been used in many video generation pipelines to improve the efficiency of training.

In T2V, Modelscope (Wang et al., 2023a) proposes spatial-temporal blocks. LaVie (Wang et al., 2023b) concatenates three latent diffusion models for base video generation and spatial and temporal super-resolution. Similarly, Show-1 (Zhang et al., 2023) concatenates three pixel-based and one latent diffusion model. VideoCrafter2 (Chen et al., 2023a; 2024) adopts a single latent diffusion model and devises a technique to better use high-quality image data. For I2V generations, SEINE (Chen et al., 2023b) designs a random masking mechanism. DynamiCrafter (Xing et al., 2023) proposes a dual-stream image injection paradigm. Both can generate transitions between two input frames. Currently, most of the open-sourced video generation models can only generate a video of 2-3 seconds in one sampling sequence. Accordingly, our benchmark features temporal transitions that could reasonably happen within a few seconds as well.

### 2.2 VIDEO GENERATION BENCHMARKS

Many large-scale text-to-video models are evaluated on the standard UCF-101 (Soomro et al., 2012) and MSRVTT (Xu et al., 2016) benchmarks by reporting FVD for video quality and CLIP similarities for text-video alignment (Radford et al., 2021). Recently, a few benchmarks and metrics have been proposed to promote more comprehensive and fine-grained video evaluation. EvalCrafter (Liu et al., 2024b) proposes a pipeline to exhaustively evaluate four aspects of the generated videos, such as text-video alignment and temporal consistency. FETV (Liu et al., 2024c) disentangles major content and attribute control in prompts to achieve a fine-grained evaluation of text-video alignment. VBench (Huang et al., 2024c) is another evaluation suite that adopts a unique evaluator for each of the 16 dimensions. T2VScore (Wu et al., 2024) uses Large Language Models (LLM) and video question answering (VQA) models to evaluate the text-video alignment. However, prompts in these benchmarks underaddress any transitions in attributes or object relations. Besides, we show that these metrics have marginal correlations with human ratings. In contrast, we are the first to design a benchmark and metrics that specifically characterize temporal compositionally.

### 2.3 COMPOSITIONALITY IN VISUAL GENERATION

Compositionality in image generation has been studied for years (Johnson et al., 2018; Yang et al., 2022; Zeng et al., 2023). Some early studies focus on learning separable latent or pixel representations for simple object generation (Andreas, 2018; Greff et al., 2019; Liu et al., 2021), while recent work studies more complex concepts and relations in open-domain image generation (Liu et al., 2022; Feng et al., 2022; Rassin et al., 2024). There are several studies on compositions in video prediction or generation. For example, (Ye et al., 2019) factories entities in an image, predict their future states and then generate future frames. AG2Vid (Bar et al., 2021) generates videos of moving blocks based on action graphs and layout inputs to achieve compositionality in time. VideoComposer (Wang et al., 2024a) uses a spatial-temporal condition encoder for sketch or motion inputs. Several other studies use LLMs to generate layouts or frame-wise text guidance (Huang et al., 2024a; Lin et al., 2023; Lian et al., 2023). Another concurrent work, VideoTetris (Tian et al., 2024), addresses multi-object scenes and long-range video transitions by applying spatial-temporal composing techniques. While more and more studies have started to address composition changes in video generation, there lacks a unified, standard, and challenging benchmark for such aspects. Previously in image generation evaluation, some metrics (Huang et al., 2024b; Saxon et al., 2024) have relied on the visual question answering (Hu et al., 2023; Singh & Zheng, 2023; Cho et al., 2023) or image captioning (Lu et al.,

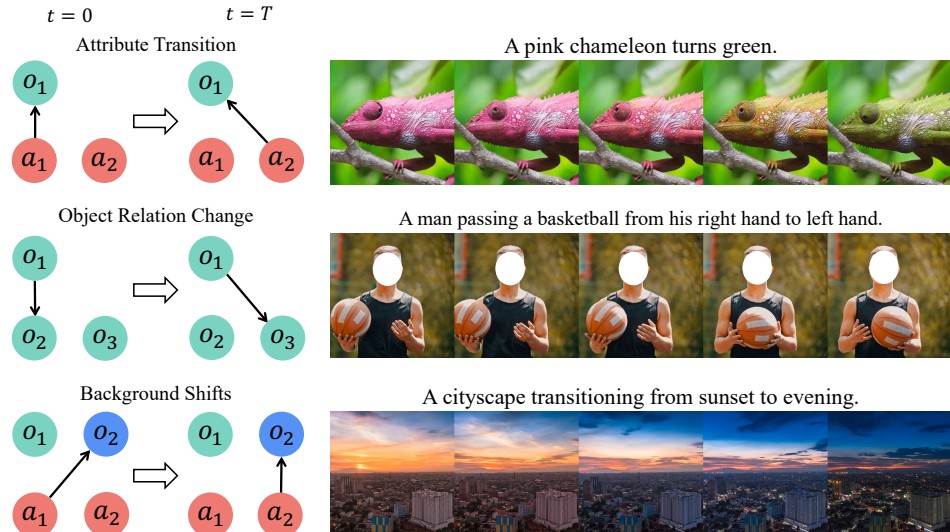

Figure 2: Three types of prompt-video pairs in TC-Bench. The left side shows the transition of video scene graphs. Green and blue nodes represent objects or scenes and red nodes represent attributes.

2024) abilities of VLMs. In this work, we rely on the image-understanding ability of VLMs to evaluate generated videos by examining video assertions on sampled keyframes.

## 3 TC-BENCH

Our Temporal Compositionality Benchmark (TC-Bench) consists of prompts following a well-defined scene graph space and ground truth videos. We first define three categories of temporal compositionality in Sec. 3.1 and then describe how we collect the samples in Sec. 3.2.

### 3.1 TEMPORAL COMPOSITIONALITY PROMPTS

Given that $o_i$ denotes an object, $a_i$ denotes an attribute, and $\rightarrow$ denotes a binding relation, $a_1 \rightarrow o_1$ means that $o_1$ has the attribute $a_1$, while $o_1 \rightarrow o_2$ means that $o_1$ and $o_2$ are interacting with each other. A scene $s_t$ at time $t$ can be represented as a combination of these elements, i.e., $s_t = \{a_1, o_1, \ldots\}$. Then, we can define three types of scenarios as shown in Fig. 2:

**Attribute Transition**: $s_0 = \{o_1, a_1 | o_1 \leftarrow a_1\} \Rightarrow s_T = \{o_1, a_2 | o_1 \leftarrow a_2\}$ means that an object's attribute changes from $a_1$ at $t = 0$ to a different one $a_2$ at the end $t = T$. A typical example is shown in Fig. 2 (top), where a chameleon's skin turns from pink to green. Prompts in this category cover a wide range of different attributes, including color, shape, material, and texture.

**Object Relation Change**: $s_0 = \{o_1, o_2, o_3 | o_1 \rightarrow o_2\} \Rightarrow s_T = \{o_1, o_2, o_3 | o_1 \rightarrow o_3\}$ indicates that an object $o_1$ interacts with different objects due to motions like passing or hitting. Fig. 2 (middle) illustrates an example where a basketball ($o_1$) is passed from the right hand ($o_2$) to the left hand ($o_3$).

**Background Shifts**: $s_0 = \{o_1, o_2, a_1 | o_2 \leftarrow a_1\} \Rightarrow s_T = \{o_1, o_2, a_2 | o_2 \leftarrow a_2\}$ is similar to attribute transition but the transition takes place on an object or scene $o_2$. $o_1$ serves as a distractor to challenge models on frame consistency while generating dynamics. For instance, in Fig. 2 bottom, the cityscape remains static while the sky changes from sunset to evening.

For simplicity, we neglect other possible nodes or edges and only focus on single transition events that could possibly happen within a short time from one second to around ten seconds.

### 3.2 DATA COLLECTION

To collect the prompts and the corresponding videos, we adopt a multi-round human-in-the-loop approach. We craft a set of video captions and verbalized type definitions. Then we feed them into

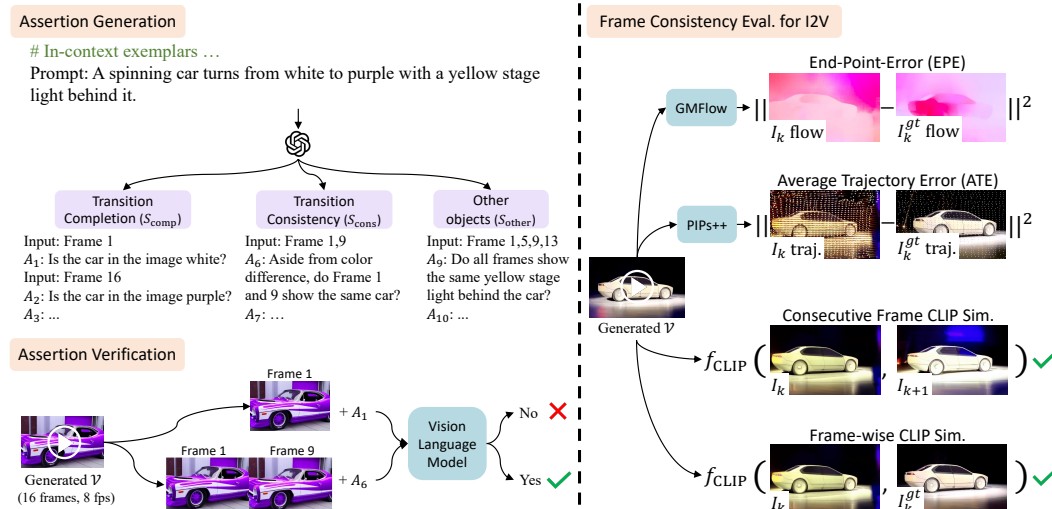

Figure 3: **Left**: Assertion generation and verification covering three evaluation dimensions. **Right**: We investigate various methods to evaluate frame consistency for I2V models and discover that CLIP-based similarities demonstrate higher correlations with human ratings.

GPT-4 and instruct it to generate more prompts following the format and definition. We manually select around 50 samples for each type, leading to TC-Bench-T2V. This set contains 150 prompts for evaluating T2V models without relying on paired videos. The prompts cover a broad spectrum of attributes, actions, and objects and explicitly depict the initial and end states of scenes to avoid any semantic ambiguity in the start and end frames.

To broaden the scope of TC-Bench, we ask human annotators to find matching videos on YouTube for the 150 prompts. If a video is highly relevant but not perfectly aligned, annotators adjust the text accordingly. Conversely, if a suitable video cannot be found, the prompt is discarded and replaced by generating new ones. We iterate over this process until we have collected 120 prompt-video pairs forming TC-Bench-I2V. The ground truth videos not only provide image inputs for I2V models but also serve as references for computing metrics. More details can be found in Appendix B.

## 4 EVALUATION METRICS

In this section, we first introduce our video assertion-based metrics to measure the text-video alignment for both T2V and I2V models (Fig. 3 left). Then, we investigate four approaches to measure frame consistency for I2V models (Fig. 3 right).

### 4.1 ASSERTION-BASED EVALUATION

Denote a text input $P$ and a video $\mathcal{V} = \{I_1, \ldots, I_K\}$ consisting of $K$ frames. We use GPT-4 to generate $N$ index-assertion pairs $\{(\mathcal{K}_i, A_i)\}$ where $\mathcal{K}_i$ consists of up to 5 different frame indices used to retrieve frames from $\mathcal{V}$ to examine the assertion $A_i$. Without constraints, generating $\mathcal{K}$ and $A_i$ simultaneously can lead to unreasonable assertions. Therefore, we indicate that $A_i$ should cover three dimensions: *transition completion* ($S_{\text{comp}}$), *transition object consistency* ($S_{\text{cons}}$), and *other objects* ($S_{\text{other}}$). We provide a few in-context exemplars so that the LLM can follow the same format. More details about these dimensions are explained in Appendix C.

To verify each assertion $A_i$, we input $A_i$ and the corresponding video frames $\mathcal{I}_i = \{I_k | k \in \mathcal{K}_i\}$ to a VLM (Achiam et al., 2023; Hong et al., 2024; Liu et al., 2024a) $f_{\text{VLM}}$. The VLM produces a response $f_{\text{VLM}}(\mathcal{I}_i, A_i) \in \{\text{Yes}, \text{No}\}$, indicating whether the assertion $A_i$ is verified. When $\mathcal{K}_i$ contains more than one index, we concatenate the frames horizontally as one image feeding into the VLM. We empirically observe that the combined image input is more reliable than sequential image inputs for TC-Bench evaluation, contrary to the findings of some recent work (Wang et al., 2024b). A transition is completed if all $A_i$ from transition completion and consistency are verified. Therefore, we define

the Transition Completion of $P$ and $\mathcal{V}$ as:

$$\text{TC}(P, \mathcal{V}) = \begin{cases} 1 & \text{if } \forall i, \mathbb{1}(f_{\text{VLM}}(\mathcal{I}_i, A_i) = \texttt{Yes}), \text{ where } A_i \in S_{\text{comp}} \cup S_{\text{cons}} \\ 0 & \text{otherwise,} \end{cases} \quad (1)$$

where $\mathbb{1}(\cdot)$ is the indicator function. It returns True when $f_{\text{VLM}}$ verifies the assertion $A_i$ according to $\mathcal{I}_i$. Therefore, we say a video $V_j$ completes the transition described by $P_j$ only when it passes all assertion $A_i \in S_{\text{comp}} \cup S_{\text{cons}}$.

To this end, we can define a model's Transition Completion Ratio (TCR) with equation 1. Given a set of $M$ text-video pair $(P_j, \mathcal{V}_j)$ generated by the model, its TCR is given as below

$$\text{TCR} = \frac{1}{M} \sum_j \text{TC}(P_j, \mathcal{V}_j) \times 100, \quad j = 1, \ldots, M. \quad (2)$$

TCR shows the percentage of videos in the whole benchmark that align with the prompts. We can further define the TC-Score of a text-video pair $(P, \mathcal{V})$ as the pass rate of all assertion examinations:

$$\text{TC-Score}(P, \mathcal{V}) = \frac{1}{N} \sum_{i=1}^{N} \mathbb{1}(f_{\text{VLM}}(\mathcal{I}_i, A_i)), A_i \in S_{\text{comp}} \cup S_{\text{cons}} \cup S_{\text{other}}, \quad (3)$$

ending up with a value within $[0, 1]$. Compared to TCR, the averaged TC-Score can be viewed as a more comprehensive metric that validates all concepts mentioned in the prompts.

## 4.2 CONSISTENCY EVALUATION FOR IMAGE-TO-VIDEO GENERATION

I2V models (Xing et al., 2023; Chen et al., 2023b), by using ground truth start and end frames as inputs, may generate adversarial intermediate frames to deceive VLMs in verifying assertion. While TCR and TC-Score still show positive correlations for these models, we find it beneficial to penalize such phenomena by evaluating frame consistency using latent features. The TC-Score for I2V models is then defined as:

$$\text{TC-Score}(P, \mathcal{V}) = w_1 \frac{1}{N} \sum_{i=1}^{N} \mathbb{1}(f_{\text{VLM}}(\mathcal{I}_i, A_i) = \texttt{Yes})) + w_2 \frac{1}{K-1} \sum_{k=1}^{K-1} f_{\text{CLIP}}(I_k, I_{\text{ref}}), \quad (4)$$

where $f_{\text{CLIP}}$ is the CLIP cosine similarity and $I_{\text{ref}}$ is either the next frame $I_{k+1}$ or the frame from the ground truth video $I_k^{\text{gt}}$. $w_1$ and $w_2$ are weighting factors. As shown in Fig. 3 (right), we explore four candidates and find that using CLIP latent features is more reliable (also see Appendix C.2).

## 5 METHOD

We introduce a simple and effective baseline to improve the transition completion rate over text-to-video generation models. Based on the prompt $P$, we first instruct an LLM to generate the text description of the initial scene $P_0$ and the end scene $P_K$. Then we utilize a diffusion-based text-to-image generation model $f_{t \rightarrow i}$ to generate the start and end frame $I_1, I_K$. However, simply using $P_0, P_K$ to guide the generation process overlooks the consistency across the frames. Therefore, we apply the same noise map $z_T$ as the initialized noise pattern of both diffusion paths and substitute the self-attention maps of $I_K$ with maps from $I_0$'s diffusion trajectory for the first half of the timesteps. As $P_0$ and $P_K$ share similar semantics except in some attributes or object positions, we discover that such a simple method can end up with $I_1$ and $I_K$ sharing similar image structures. Then, the generated frames are used to guide the process of video generation so that the temporal transition can be completed under the guidance of $I_k$. We use an off-the-shelf video generation model SEINE (Chen et al., 2023b) for the generative transition from $I_1$ to $I_K$. We refer to this baseline as *SDXL+SEINE* as we adopt SDXL (Podell et al., 2023) for start and end frame generation.

## 6 EXPERIMENT

### 6.1 EXPERIMENT SETUP

**Baselines** Including the above SEINE-based method, we consider fourteen T2V models/systems across three major types and two I2V models that perform generative frame interpolation. We

Table 1: Automatic evaluation results of three types of baselines on TC-Bench-T2V. The bold text highlights the best metric scores in each type of method. Multi-stage T2V methods adopt LLMs or text-to-image models to generate additional conditions for video generation.

| | | TC-Bench-T2V | | | | | | | |
| --- | --- | --- | --- | --- | --- | --- | --- | --- | --- |
| | | Attribute | | Object | | Background | | Overall | |
| | Methods | TCR | TC-Score | TCR | TC-Score | TCR | TC-Score | TCR ↑ | TC-Score ↑ |
| | *Open-source models: Text → Video* | | | | | | | | |
| 1 | ModelScope (Wang et al., 2023a) | 3.52 | 0.5942 | 4.72 | 0.6230 | 3.54 | 0.5715 | 3.90 | 0.5955 |
| 2 | Show-1 (Zhang et al., 2023) | 3.85 | 0.6029 | 5.58 | 0.6544 | 5.49 | 0.6008 | 4.95 | 0.6182 |
| 3 | Open-Sora-Plan v1.2 (Lab & etc., 2024) | 5.77 | 0.6241 | 2.98 | 0.6764 | 2.75 | 0.5359 | 3.87 | 0.6105 |
| 4 | Open-Sora v1.2 (hpcaitech, 2024) | 6.15 | 0.6509 | 7.66 | **0.7406** | 2.35 | 0.5847 | 5.33 | 0.6565 |
| 5 | LaVie (Wang et al., 2023b) | 4.63 | 0.5807 | 6.06 | 0.6323 | 6.28 | 0.6252 | 5.64 | 0.6119 |
| 6 | VideoCrafter2 (Chen et al., 2024) | 4.25 | 0.6166 | 6.44 | 0.6724 | **7.06** | **0.6338** | 5.89 | 0.6399 |
| 7 | CogVideoX-5B (Yang et al., 2024) | **8.08** | **0.6930** | **10.64** | 0.7237 | 4.71 | **0.6338** | **7.73** | **0.6825** |
| | *Proprietary models/systems: Text → Video* | | | | | | | | |
| 8 | Pika 1.0 (Pik, 2023) | 5.77 | 0.6520 | 8.51 | 0.7242 | 1.96 | 0.6070 | 5.33 | 0.6593 |
| 9 | Kling 1.0 (Kli, 2024) | 7.69 | 0.6888 | 10.64 | **0.7819** | 3.92 | 0.6183 | 7.33 | 0.6940 |
| 10 | Dream Machine (Lum, 2024) | **9.80** | 0.7319 | **12.77** | 0.7755 | 5.88 | 0.6284 | 9.40 | 0.7102 |
| 11 | Gen-3 Alpha (Gen, 2024) | 9.62 | **0.7507** | 10.64 | 0.7073 | **27.45** | **0.7488** | **16.00** | **0.7365** |
| | *Multi-stage T2V: Text → Text/Layout/Images → Video* | | | | | | | | |
| 12 | Free-Bloom (Huang et al., 2024a) | 6.32 | 0.6256 | 6.84 | 0.6215 | 24.02 | 0.7394 | 12.55 | 0.6633 |
| 13 | LVD Lian et al. (2023) | 5.77 | 0.6215 | **12.77** | **0.7081** | 1.96 | 0.5042 | 6.67 | 0.6088 |
| 14 | SDXL+SEINE (Ours) | **13.08** | **0.6579** | 5.60 | 0.6486 | **35.43** | **0.7916** | **18.37** | **0.6993** |

Table 2: Automatic evaluation results of I2V models on TC-Bench-I2V.

| | | TC-Bench-I2V | | | | | | | |
| --- | --- | --- | --- | --- | --- | --- | --- | --- | --- |
| | | Attribute | | Object | | Background | | Overall | |
| | Methods | TCR | TC-Score | TCR | TC-Score | TCR | TC-Score | TCR ↑ | TC-Score ↑ |
| | *Start & End Frame → Video* | | | | | | | | |
| 15 | SEINE | **17.86** | 0.7197 | 10.48 | 0.6541 | 7.96 | 0.7421 | 13.57 | 0.6978 |
| 16 | DynamiCrafter | 16.55 | **0.7449** | **13.91** | **0.7074** | **25.56** | **0.7949** | **16.89** | **0.7380** |

benchmark major open-source T2V models, such as *VideoCrafter2* (VC2) (Chen et al., 2023a; 2024) and *CogVideoX-5B* (Yang et al., 2024), most recent proprietary systems such as Kling and Gen-3 Alpha, and finally, multi-stage T2V models such as Free-Bloom (Huang et al., 2024a) and LVD (Lian et al., 2023). *Free-Bloom* (Huang et al., 2024a) applies an LLM to generate a list of prompts that are used to guide generation for different frames. We re-implement it on top of VideoCrafter2 for optimal results. *LVD* (Lian et al., 2023) applies an LLM to generate bounding boxes for each frame and synthesize videos with a layout-to-video model. For I2V models, *SEINE* (Chen et al., 2023b) and *DynamiCrafter* (Xing et al., 2023) take the first and last frames from ground truth videos and generate intermediate frames. Additional implementation details are clarified in Appendix A.

**Metrics** For the proposed *TCR* and *TC-Score*, we adopt GPT-4 Turbo, CogVLM2-19B, and LLaVA-NeXT-7B to assess all the assertions. The results reported in the main paper are based on GPT-4 Turbo, and the other results are reported in Appendix A and Table 6 and 7. In comparison, we consider four commonly used or recent text-video alignment metrics. *CLIP score* (Radford et al., 2021) measures the average text-frame similarity. *ViCLIP* (Wang et al., 2023c) encodes video and text as two separate feature vectors, which can be used to compute cosine similarity as the text-video alignment score (Huang et al., 2024c). *EvalCrafter* (Liu et al., 2024b) computes a weighted sum of many different metrics, but we only adopt the sum of CLIP score, SD score, and BLIP-BLEU since these metrics are agnostic to prompt content and structure. Finally, *UMTScore* (Liu et al., 2024c) uses the video-text matching score from a fine-tuned UMT (Li et al., 2023b), an advanced video foundation model. We also collect human ratings with a 5-point Likert scale to compute correlations with these automatic metrics.

## 6.2 QUANTITATIVE RESULTS

**Direct T2V Models** Table 1 shows the automatic evaluation results of T2V baselines on three types of scenarios of TC-Bench. The overall TCR and TC-Score indicate a clear discrepancy between open-source and proprietary models, except that CogVideoX-5B (Yang et al., 2024) achieves similar

performances as Kling. CogVideoX-5B is particularly strong in showing object relation changes. Gen-3 Alpha demonstrates apparent superiority in background transitions with a 27.45% TCR, while Dream Machine achieves the best results in object relation transitions with a 12.77% TCR. As no single model/system dominates over all three types of prompts, we conjecture that the results implicitly reflect the differences in data curation between different models/systems. The overall results also evidence the difficulty of TC-Bench prompts, even though they are mostly single-hop transitions in concept. Our rank of T2V models also aligns with established benchmarks such as VBench (Huang et al., 2024c) or EvalCrafter (Liu et al., 2024b).

**Multi-stage T2V models**, including Free-Bloom (Huang et al., 2024a), LVD (Lian et al., 2023), and our SDXL+SEINE, generates frame-wise prompts, layouts, and images as intermediate steps. While these methods effectively enhance the overall TCR, the unbalanced fluctuations across types reveal limitations using explicit mid-level representations. For instance, LVD fails to address attributes or backgrounds because these transitions cannot be represented using bounding boxes. SDXL+SEINE underperforms in object relation because T2I models struggle to control object positions in two diffusion paths of similar structures. The results suggest the necessity of fundamentally addressing the gap between video and text features in the latent space to tackle TC-Bench.

**I2V Models**  As shown in Table 2, SEINE (Chen et al., 2023b) and DynamiCrafter (Xing et al., 2023) achieve much higher TCR than T2V models because they are designed for transition completion. Both achieve high TCR in attribute and background as these types usually involve fewer temporal dynamics. However, as is shown later in Sec. 6.3 & 6.4, the major challenge for generative frame interpolation is to maintain frame consistency and smoothness, especially between the conditional (start and end) frames and neighboring frames. We observe that both models are still weak in maintaining consistency and coherence when the discrepancy between the start and end frames is significant.

## 6.3 QUALITATIVE RESULTS

We show several representative examples in Fig. 4. For attribute binding, a common phenomenon is that direct T2V models blend multiple concepts that should appear in different timesteps as a static pattern throughout the video (first row) or display one dominant attribute (second row). In contrast, SDXL+SEINE can generate color changes gradually. Object relation changes are more challenging, yet some of the best open-source and commercial models/systems, such as Open-Sora, CogVideoX, and Kling, can synthesize the process. While Dream Machine and Gen-3 Alpha fail in this example, we show their success cases in the Appendix. Lastly, Gen-3 Alpha, Free-Bloom, and SDXL+SEINE show strong results in background shifts of the cityscape. Still, the latter two are weaker in maintaining transition smoothness and scene consistency, resulting in lower TC-Score as in Table 1.

## 6.4 ANALYSIS

**Temporal Compositionality**  We demonstrate temporal compositionality in the generated videos by visualizing the existence of attributes $a_1, a_2$ at different time steps. Specifically, we compute the CLIP similarity between each frame and captions "a $a_1$ $o_1$" to obtain Fig. 5 (a), and captions "a $a_2$ $o_1$" to obtain Fig. 5 (b). For instance, if the video prompt is "a pink chameleon turns green", then the two captions are "a pink chameleon" and "a green chameleon" respectively. The similarity to $a_1$ should decrease while the similarity to $a_2$ should increase as the frame index increases. The flat curves of T2V models indicate that they fail to generate the disappearance of $a_1$ or the emergence of $a_2$ as time proceeds, which aligns with the case in Fig. 4. In contrast, SEINE and DynamiCrafter align well with the trend of ground truth videos.

**Frame Consistency**  Despite the fact that I2V models align with the trend of ground truth videos in Fig. 5 (a)-(b), they suffer from more severe consistency issues than T2V models. Fig. 5 (c) shows the CLIP similarity between two consecutive frames. I2V models are generally weaker than T2V models in frame consistency, especially the consistency between the start and end frames (input conditions) and their neighboring frames (model outputs). For T2V models, chasing higher consistency scores does not help achieve temporal compositionality, as most transitions cannot be completed. Therefore, we argue that it is only necessary to compute consistency for I2V models as in Eq. 4.

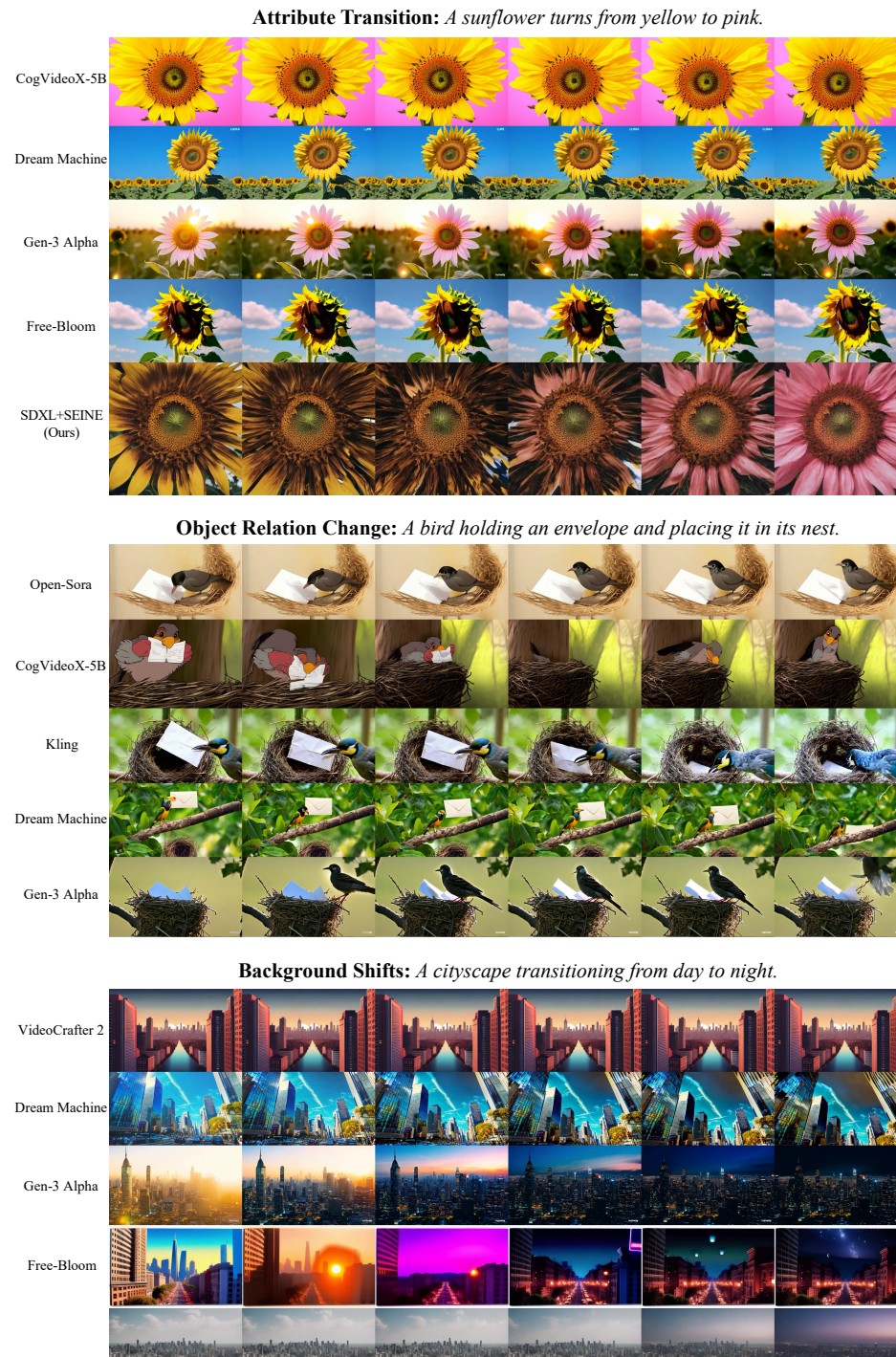

Figure 4: Qualitative comparison between different models in attribute and object binding transitions.

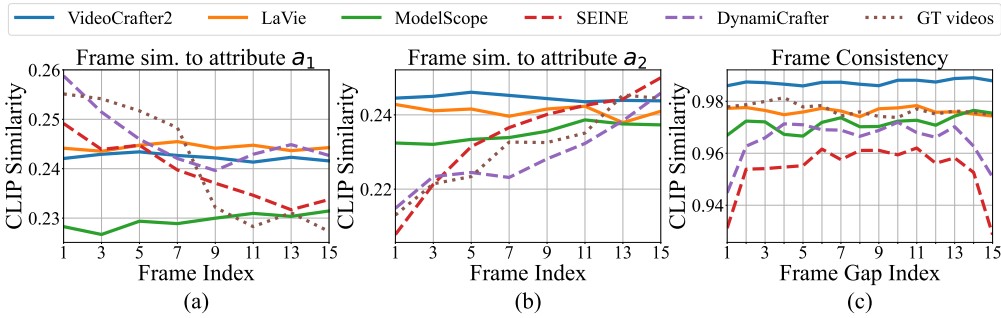

Figure 5: **(a)** Averaged CLIP cosine similarity between frame $I_k$ and the start attribute $a_1$. **(b)** Averaged CLIP cosine similarity between frame $I_k$ and end attribute $a_2$. (a) and (b) reflect the existence of $a_1, a_2$ as time proceeds. **(c)** CLIP cosine similarity between two consecutive frames.

Table 3: Correlations between human annotations and automatic evaluation metrics. The last row refers to the averaged correlation between two different annotators to show that the ratings are consistent across individuals.

|  | TC-Bench-I2V | | | |
|  | Q1: Transition Completion | | Q2: Overall Text-Video Alignment | |
| **Metrics** | Spearman $\rho$ | Kendall's $\tau$ | Spearman $\rho$ | Kendall's $\tau$ |
| --- | --- | --- | --- | --- |
| CLIP Sim. (Radford et al., 2021) | -0.0879 | -0.1211 | -0.0927 | -0.1273 |
| ViCLIP (Huang et al., 2024c) | 0.0599 | 0.0760 | 0.0465 | 0.0660 |
| EvalCrafter (Liu et al., 2024b) | 0.1098 | 0.1515 | 0.1045 | 0.1468 |
| UMTScore (Liu et al., 2024c) | 0.1508 | 0.2074 | 0.1927 | 0.2659 |
| TC-Score (Ours) | **0.2977** | **0.3753** | **0.4513** | **0.5913** |
| Human (Upper bound) | 0.7011 | 0.7724 | 0.6735 | 0.7289 |

## 6.5 HUMAN EVALUATION

We compute Kendall and Spearman's rank correlations to show that our proposed metrics align with human judgments. We collect two ratings for each video where the first one only considers transition completion and the other one considers overall text-video alignment (details in Appendix D). As is shown in Table 3, our metrics achieve much higher correlations compared to existing metrics in both aspects. The results verify the effectiveness of our metrics for evaluating temporal compositionality. Despite being widely adopted in existing studies, averaged text-frame CLIP similarity is unreliable and often outputs low scores for videos that complete the transitions. The results are intuitive as the training text samples for CLIP describe static images instead of transitions or motions, lacking awareness of compositional change across timesteps. In addition, we find that advanced text-video alignment models like ViCLIP and UMTScore are still weak in understanding temporal compositionality, leading to low correlations.

## 7 CONCLUSION

In this work, we propose a new video generation benchmark TC-Bench, featuring temporal compositionality. TC-Bench characterizes three different types and a wide range of topics. We show that simple transitions that can happen in several seconds remain extremely challenging to existing T2V methods. We also propose assertion-based evaluation metrics and investigate consistency evaluation using flow-based methods or latent features. Our benchmark, experimental results, and analysis unveil the weaknesses of existing T2V and I2V models in temporal compositionality, suggesting crucial directions for future improvement. Future work should investigate techniques to 1) automatically mine videos with specific temporal compositionality and generate detailed captions, 2) evaluate text-video alignment more efficiently, and 3) improve text-to-video models in addressing temporal compositionality.

## 8 REPRODUCIBILITY STATEMENT

We upload the benchmark prompts and video URLs in Sec. 3, generated assertions and evaluation results in Sec. 4 and 6.2 to the supplementary materials for reproducibility. We will release the full benchmark, evaluation scripts, and results. In addition to referring to the materials, we have disclosed implementation details in 6.1 and Appendix A.

## 9 ETHICS STATEMENT

For the human evaluation in Sec. 6.5, we use the Amazon Mechanical Turk platform and form the comparison task as batches of HITs. We recruit a small group of annotators who are native English speakers since the task requires understanding the English input prompt. Each HIT takes around 15-30 seconds on average to accomplish, and we pay each submitted HIT with 0.3 US dollars, resulting in an hourly payment of 36 US dollars. We will release the dataset, including the prompts, video URLs, downloading scripts, pre-generated assertions, our evaluation results, and generated videos.

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

## A  IMPLEMENTATION DETAILS

We generate five videos per prompt per model which ends up with 750 videos in total. We set the fps to 8 and the total number of frames to 16 per video, except that Show-1 is fixed to 29 frames. We use the default resolution of each model, i.e., $320 \times 512$ for VideoCrafter2, LaVie, and DynamiCrafter, $256 \times 256$ for ModelScope and LVD, $320 \times 576$ for Show-1, and $512 \times 512$ for SEINE. We use GPT-4-turbo API for frame index and assertion generation and GPT-4V for TCR and TC-Score evaluation for all videos and all models. As for $f_{\text{CLIP}}$ in Eq. 4, we first apply CLIP ViT-L/14@336px to extract frame features as a vector and compute the cosine similarity between two normalized feature vectors. We heuristically set the range of acceptable similarity scores as $[0.90, 0.98]$ based on the minimum and maximum values of ground truth videos. Scores within this range are linearly mapped to values between $[0, 1]$. Scores outside this range are adjusted accordingly: values below 0.90 are set to 0, and values above 0.98 are set to 1. We heuristically set $w_1$ to $\frac{2}{3}$ and $w_2$ to $\frac{1}{3}$ since consistency is one of the total three evaluation dimensions. We use the "Consecutive Frame CLIP Sim." because it demonstrates the highest ranking correlations with human ratings, as shown in Table

4 in Appendix C. All models can be run on a single 40 GB NVIDIA A100, and the evaluation is conducted through OpenAI API calls.

For all videos generated from open-source models, we use the default parameters (including fps, resolution, and number of frames) from the official GitHub repository. For Open-Sora-Plan v1.2, we generate 93 frames under 480p, and for Open-Sora v1.2, we generate 2-second videos. For commercial models, we generate one video for each prompt and use the default settings from the GUI. For Kling, we generate 5-second videos in professional mode. For Dream Machine, we enabled the prompt enhancement. For Gen-3 Alpha, we generate 5-second videos.

## B TC-BENCH DATASET CONSTRUCTION

This section provides the details of the prompts for generating prompts in TC-Bench using ChatGPT. We start with general instructions on the desired structure and format of temporal compositionality prompts, followed by several manually written examples. The text prompts and metadata of TC-Bench are available at this link and also at the project website.

**Attribute Transition** We explicitly ask ChatGPT to imagine scenarios where the attribute (including lighting, color, material, shape, and texture) of a certain object changes and then generates the corresponding prompts. *Generate some concise prompts that describe scenarios where an object's attribute, such as lighting, color, material, shape, or texture, changes as time proceeds. The prompt should describe transitions that could happen within a few seconds in a video. The described transition should also be realistic and could happen in the real world. Here are some examples:*

*A chameleon's skin changes from brown to bright green.*

*A leaf changing color from vibrant green to rich autumn red.*

*A car transitioning from silver to matte black.*

**Object Relation Change** We first describe the idea of object binding and then instruct ChatGPT to generate prompts that describe transitions in the binding relations. We also prompt ChatGPT to consider many different subjects as it is biased towards mentioning human occupations. *Generate some concise prompts that describe scenarios where objects' binding relations change due to some actions or motions. Two objects are bound to each other if they are physically interacting with each other. For example, in "a man passes a ball from left hand to right hand" the ball is bound to the man's left hand at first. Then, the binding relation changes from ball and left hand to ball and right hand. The prompt should describe motions that could happen within a few seconds in a video. Consider a wide range of subjects not limited to humans or one's occupation, such as animals or common objects. Here are more examples:*

*A man picking an apple from a tree and placing it in a basket.*

*A bird picking up a twig and placing it in its nest.*

*A child placing a toy car on a toy track.*

**Background Shifts** is similar to attribute transition in prompting. The major difference is that we clarify that the transition takes place on a background scene or object, with a foreground object serving as the distractor. *Generate some concise prompts that describe scenarios where a foreground object remains relatively static and the background changes as time proceeds. The prompt should describe transitions that could happen within a few seconds in a video, whether it is a normal-speed video or a timelapse video. Here are some examples:*

*A cityscape transitioning from day to night.*

*A forest changing from summer greenery to autumn foliage.*

*A bench by a lake from foggy morning to sunny afternoon.*

To ensure the integrity and quality of the data collection process, contributors must possess a nuanced understanding of temporal compositionality and the dynamics of scene graph transitions, as depicted in Figure 2. Given these specialized requirements, we opted to engage a team of students who have

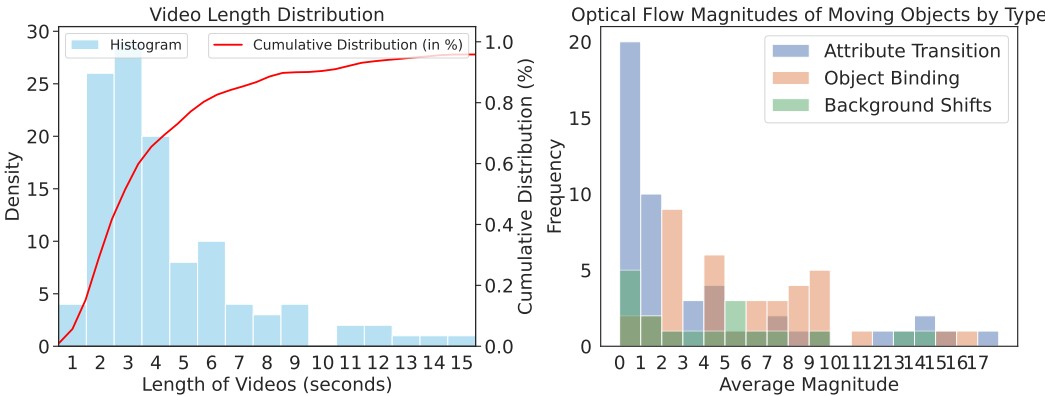

Figure 6: **Left**: Length distribution of ground truth videos. **Right**: Distribution of dynamics degree of moving object in ground truth videos.

a background in the relevant domain. Crowdsource workers, while effective for broad-range tasks, may not possess the domain-specific knowledge or the detailed task familiarity necessary for this particular study.

### B.1  GROUND TRUTH VIDEO COLLECTION AND STATISTICS

After obtaining a certain number of prompts for each type, each annotator manually searches YouTube for videos that match or are relevant to the prompts. If a video illustrates temporal compositionality but does not fully align with the prompt, annotators will revise the prompt to align with the video instead. If relevant videos cannot be found after several search trials, we discard the prompt and proceed to the next one. The annotators record the YouTube ID, start time, and end time for each video. This metadata is shared with the users of TC-Bench for downloading ground truth videos. We also ensure that the video length is within a reasonable range from several seconds to less than 20 seconds.

Fig. 6 provides two collected video statistics. On the left, we show the distribution of the video lengths to prove that the events described in our prompts are realistic and could happen within a few seconds. Note that around 80% of the videos have a length shorter than or equal to 6 seconds, and 95% of the videos are shorter than 15 seconds. On the right, we show the distribution of dynamic degrees of all videos using optical flow. We first extract the optical flow for each frame and compute the flow magnitude of each pixel. Then we apply a threshold to eliminate static background area and compute the average magnitude over the remaining area that are moving objects or areas. We observe that videos from attribute transition and background shifts contain less motion than those from object binding changes. This aligns with our intuition because the latter often needs human actions or subject motion to accomplish compositional change.

## C  EVALUATION METRICS

This section provides more details about assertion generation and frame consistency evaluation for I2V models.

### C.1  ASSERTION GENERATION

As described in Sec. 4 and Fig. 3, we provide three in-context exemplars for GPT-4 to generate assertions for each prompt from TC-Bench-T2V and TC-Bench-I2V. We manually write one exemplar for each type and append them after the instruction. The detailed prompt is shown in Table 8 and 9. The three dimensions are *transition completion*, *transition consistency*, and *other objects*. Transition completion first checks whether the start and end frames reflect the required concepts. To detect unnatural videos with abrupt changes between two consecutive frames, assertions also check an intermediate frame and a sequence of sampled frames. Transition consistency further examines

whether the objects in intermediate frames maintain key identity features as in the first frame. Finally, we also check for other objects beyond those mentioned in the prompt, such as the distractor object "bench" in Table 9.

## C.2 FRAME CONSISTENCY FOR I2V MODELS

As is introduced in Sec. 4 and Fig. 3, we investigate four different methods to measure consistency for generative frame interpolation. Note that since the ground truth videos are in arbitrary length and an arbitrary number of frames, we first sample 16 frames with equal gaps from each video to match the number of frames in the generated videos. Then, we apply different methods to extract optical flow, trajectory, or latent features.

- **End-Point-Error (EPE)** is a standard metric from optical flow estimation that measures the Euclidean distance between the vectors from two optical flow maps. We first use GMFlow Xu et al. (2022) to extract optical flow vectors $(u_k, v_k)$ for each pixel in frame $k$ in the generated videos. For simplicity, we omit the pixel index hery. Then we also extract $(u_k^{\text{ref}}, v_k^{\text{ref}})$ from the ground truth videos. End-Point-Error is simply an averaged $L_2$ distance between every pair of optical flow vectors of all pixels in all frames:

$$EPE = \frac{1}{|\mathcal{P}|} \sum_{p \in \mathcal{P}} \frac{1}{K} \sum_k \sqrt{(u_k - u_k^{\text{ref}})^2 + (v_k - v_k^{\text{ref}})^2}, \quad (5)$$

  where $\mathcal{P}$ in the set of pixels in a frame and $p \in \mathcal{P}$ represents all pixels within the frame.

- **Average Trajectory Error (ATE)** is a standard measure used in point tracking in video sequences or other dynamic contexts. It quantifies the average discrepancy between the estimated trajectories of points and their ground truth trajectories over time. We estimate the position of 1024 points $\hat{p}_k \in \mathbb{R}^2$ for each frame and the reference $p_k \in \mathbb{R}^2$ from ground truth videos. The ATE is the averaged position differences over all $K$ frames:

$$ATE = \frac{1}{|\mathcal{P}|} \sum_{p \in \mathcal{P}} \frac{1}{K} \sum_k \|p_k - \hat{p}_k\|_2. \quad (6)$$

- **Frame Consistency Error** (i.e. Consecutive Frame CLIP Sim. in Fig. 3), introduced in Esser et al. (2023), is to compute the cosine similarity between features of two consecutive frames extracted by CLIP Image encoder.

- **Frame-wise CLIP Similarity** is to compute the cosine similarity between features of the generated frame and corresponding ground truth frames.

Since these metrics are investigated to measure consistency, we process the collected human ratings to disentangle the score sets from involving transition completion consideration. In our 5-point Likert scale, a score of 4 indicates that the transition is completed, but there are consistency issues. A score of 5 indicates that the transition is completed and there are merely consistency issues. Since each video has three different ratings, we filter out videos with an average score below 3.6 to ensure that each has at least two scores of 4 or 5. This has led to 128 videos from I2V models. However, for T2V models, the completion rate is too low that over 97% of the videos have average scores below 3. We are unable to disentangle consistency from transition completion for T2V models. This is another reason we only accommodate frame consistency error for I2V models as stated in Sec. 4 and Eq. 4.

Table 4 presents the ranking correlations between these four metrics and processed human ratings. Consecutive Frame CLIP Similarity achieves the highest correlation scores and is unsupervised. We conjecture that EPE and ATE are too strict for TC-Bench evaluation because there can be many possible ways to generate natural transitions between two frames. We indeed observe cases where the generated video contains a huge amount of dynamics and completes the attribute transition smoothly. However, the ground truth video shows a static object changing attributes. Such discrepancy could have caused misalignments between the automatic scores and human ratings.

## D  HUMAN EVALUATION

We first generate five videos per prompt per model for human annotations using Modelscope, LaVie, VC2, SEINE, and DynamiCrafter on TC-Bench-I2V. This is to unify the prompt space for T2V and

Table 4: Ranking correlations between frame consistency measurements and processed human ratings for SEINE and DynamiCrafter.

| Metrics | Unsupervised | Transition Completion Ratings | |
| --- | --- | --- | --- |
| | | Spearman $\rho$ | Kendall's $\tau$ |
| End-Point-Error | ✗ | -0.1742 | -0.2320 |
| Average Trajectory Error | ✗ | -0.1579 | -0.2149 |
| Frame-wise CLIP Sim. | ✗ | 0.2326 | 0.3107 |
| Consecutive Frame CLIP Sim. | ✓ | 0.2861 | 0.3807 |

Table 5: Automatic and human evaluation results of T2V and I2V models on TC-Bench-I2V. The results are used to compute ranking correlations.

| | Methods | TC-bench-I2V | | | | | | | | Human Ratings | | |
| --- | --- | --- | --- | --- | --- | --- | --- | --- | --- | --- | --- | --- |
| | | Attribute | | Object | | Background | | Overall | | Completion rate Q1¿=3.66 | Q1 ratings | Q2 ratings |
| | | TCR | TC-Score | TCR | TC-Score | TCR | TC-Score | TCR ↑ | TC-Score ↑ | | | |
| | *Text → Video* | | | | | | | | | | | |
| 1 | ModelScope | 4.76 | 0.5577 | 1.33 | 0.5604 | 4.17 | 0.5330 | 3.28 | 0.556 | 0.00 | 1.304 | 1.727 |
| 2 | LaVie | 1.30 | 0.5329 | 1.33 | 0.5399 | 10.71 | 0.5967 | 2.78 | 0.5457 | 0.55 | 1.357 | 1.726 |
| 3 | VideoCrafter | 3.45 | 0.6187 | 12.33 | 0.6304 | 11.11 | 0.6898 | 8.02 | 0.6335 | 1.07 | 1.344 | 1.840 |
| | *Start & End Frame → Video* | | | | | | | | | | | |
| 4 | SEINE | **17.86** | 0.7197 | 10.48 | 0.6541 | 7.96 | 0.7421 | 13.57 | 0.6978 | 22.56 | 2.895 | 2.837 |
| 5 | DynamiCrafter | 16.55 | **0.7449** | **13.91** | **0.7074** | **25.56** | **0.7949** | **16.89** | **0.7380** | **27.82** | **2.980** | **2.970** |

I2V models to reduce bias during annotation. Then, we randomly sampled around 900 videos from all the videos and assigned three different annotators for each video to reduce variance. We discarded the videos with divisive ratings and ended up with 2451 human ratings over 817 generated videos. The detailed graphical user interface for rating collection is shown in Fig. 7. We design two questions, the first focusing on transition only while the second considering the overall text-video alignment in favor of measuring the transition. We release these ratings along with the benchmark data and metrics for future work to improve the evaluation protocols further.

This data annotation part of our project is classified as exempt by the Human Subject Committee via IRB protocols. We launched our annotation jobs (also called HITs) on the Amazon Mechanical Turk platform. We recruited eight native English-speaking workers and provided thorough instructions and guidance to help them understand the task's purpose and the emphasis of each question. We also provided five detailed examples in the annotation interface for their reference and communicated with the workers to resolve confusion throughout the process. The workers' submissions are all anonymous, and we did not collect or disclose any personally identifiable information in the collection stage or dataset release. Our prompts and generated videos do not contain offensive content. Each HIT has a reward of 0.35 USD and takes around 40 seconds to complete, leading to an hourly rate of 31.5 USD and a total cost of 1134 USD.

# E ADDITIONAL RESULTS

**Quantitative Results** We show the complete results of TC-bench-I2V in Table 5 with human evaluation. We calculate the ratio of videos with a Q1 rating larger than 3.66 to extract a measurement from human ratings with similar meanings to TCR. However, note that this measurement is not statistically the same as TCR, and its value cannot be directly compared with TCR. It is designed to reflect the overall ranking of models in terms of transition completion. I2V models achieve a much higher completion rate than T2V models, which only achieve around 1%. The low average ratings in Q1 and Q2 also imply the lack of temporal compositionality in existing T2V models.

**Qualitative Results** We show additional qualitative comparisons of baselines in Fig. 8 - 13. Compared to direct T2V models or multi-stage T2V models, our SDXL+SEINE achieves better temporal compositionality by showing more significant transitions in Fig. 8 & 10. However, as is shown in Fig. 9, it still suffers from generating dynamics for object relation change. The intermediate frames also show consistency issues. While LVD demonstrates the correct dynamics, it suffers from low visual quality and consistency issues as well.

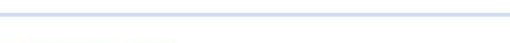

**Read a prompt, watch a generated videos, and then rate the video.**

Instruction: You are provided with a text prompt describing transitions in attributes (like colors or materials), or changes in object location or motion. Below is a generated video based on this prompt.

- For Q1, rate the video regarding **whether the described transition is represented in the video**.
- For Q2, rate the video based on the **overall video-prompt alignment**.
  Overall alignment considers both whether the transition is completed and whether all mentioned objects are correctly shown in the video. With the former slightly outweights the latter.
  See the examples below.

> Examples (click to expand)                                                    ⌄

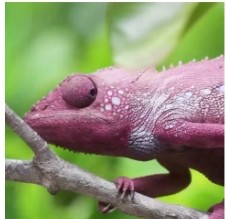

**Prompt: A pink chameleon turns green**

**Q1: Rate how good the video demonstrates the transition described in the prompt.**
1 - Very bad. No evidence of the described transition.
2 - Bad. A slight hint of the transition, but barely noticeable or recognizable.
3 - Neutral. Transition is noticeable but not fully realized or very unnatural.
4 - Good. Transition is complete, but the process is somewhat unclear or illogical.
5 - Very good. Transition is complete and the process is smooth and natural.

**Q2: Rate how good is the overall video-prompt alignment.**
1 - Very bad. No transitions + incorrect or missing objects.
2 - Bad. No transition + correct objects *OR* Partial/Unnatural transition + missing/incorrect objects.
3 - Neutral. Partial/Unnatural transition + correct objects.
4 - Good. Complete and smooth transition + other objects are missing/incorrect.
5 - Very good. Complete and smooth transition + all objects correctly depicted.

Figure 7: Screenshot of our job on Amazon Mechanical Turk to collect human ratings for generated videos.

Table 6: Automatic evaluation results of three types of baselines on TC-Bench-T2V using `llava-v1.6-mistral-7b-hf`. The overall TCR ranks exhibit a correlation coefficient of 0.8569 with the evaluation results using GPT-4 Turbo, while TC-Score ranks demonstrate a correlation coefficient of 0.8643.

| | | TC-Bench-T2V | | | | | | | |
| --- | --- | --- | --- | --- | --- | --- | --- | --- | --- |
| | | Attribute | | Object | | Background | | Overall | |
| | Methods | TCR | TC-Score | TCR | TC-Score | TCR | TC-Score | TCR ↑ | TC-Score ↑ |
| | Open-source models: *Text → Video* | | | | | | | | |
| 1 | ModelScope (Wang et al., 2023a) | 32.69 | 0.8465 | 38.30 | 0.8319 | 29.80 | 0.7939 | 33.47 | 0.8240 |
| 2 | Show-1 (Zhang et al., 2023) | 41.92 | 0.8707 | 48.94 | 0.8791 | 29.41 | 0.8116 | 39.87 | 0.8532 |
| 3 | Open-Sora-Plan v1.2 (Lab & etc., 2024) | 30.77 | 0.8208 | 29.79 | 0.7813 | 30.98 | 0.7930 | 30.53 | 0.7990 |
| 4 | Open-Sora v1.2 (hpcaitech, 2024) | 38.08 | 0.8322 | 46.38 | 0.8596 | 30.59 | 0.8152 | 38.13 | 0.8350 |
| 5 | LaVie (Wang et al., 2023b) | 35.39 | 0.8547 | 40.85 | 0.8418 | 34.51 | 0.8255 | 36.80 | 0.8407 |
| 6 | VideoCrafter2 (Chen et al., 2024) | 36.54 | 0.8595 | 47.66 | 0.8958 | 40.78 | 0.8611 | 41.47 | 0.8714 |
| 7 | CogVideoX-5B (Yang et al., 2024) | 45.39 | 0.8866 | 47.23 | 0.8768 | 33.73 | 0.8316 | 42.00 | 0.8649 |
| | Proprietary models/systems: *Text → Video* | | | | | | | | |
| 8 | Pika 1.0 (Pik, 2023) | 32.69 | 0.8625 | 53.19 | 0.8819 | 35.29 | 0.8653 | 40.00 | 0.8695 |
| 9 | Kling 1.0 (Kli, 2024) | 42.31 | 0.8792 | 61.70 | 0.9301 | 39.22 | 0.8523 | 47.33 | 0.8860 |
| 10 | Dream Machine (Lum, 2024) | 52.94 | 0.8932 | 72.34 | 0.9451 | 15.69 | 0.7683 | 46.31 | 0.8668 |
| 11 | Gen-3 Alpha (Gen, 2024) | 50.00 | 0.9080 | 40.43 | 0.8578 | 58.82 | 0.8990 | 50.00 | 0.8892 |
| | Multi-stage T2V: *Text → Text/Layout/Images → Video* | | | | | | | | |
| 12 | Free-Bloom (Huang et al., 2024a) | 57.31 | 0.8930 | 33.62 | 0.7993 | 49.80 | 0.8645 | 47.33 | 0.8539 |
| 13 | LVD Lian et al. (2023) | 23.08 | 0.7798 | 27.66 | 0.7519 | 19.61 | 0.7218 | 23.33 | 0.7513 |
| 14 | SDXL+SEINE (Ours) | 62.69 | 0.9177 | 51.92 | 0.9121 | 63.92 | 0.9196 | 59.73 | 0.9166 |

Fig. 11-13 shows direct comparison between T2V models and I2V models. The main issue of T2V models is that they cannot generate different semantics in different frames, as described in the prompts. T2V models mix up a group of concepts and visualize them simultaneously in each frame

Table 7: Partial evaluation results of three types of baselines on TC-Bench-T2V using `cogvlm2-llama3-chat-19B`. Due to computational resource limitations, we only run five models. The overall TCR ranks align with the evaluation results using GPT-4 Turbo, while TC-Score ranks demonstrate a correlation coefficient of 0.9000.

| | | TC-Bench-T2V | | | | | | | |
| | | Attribute | | Object | | Background | | Overall | |
| | Methods | TCR | TC-Score | TCR | TC-Score | TCR | TC-Score | TCR ↑ | TC-Score ↑ |
|---|---|---|---|---|---|---|---|---|---|
| 7 | CogVideoX-5B (Yang et al., 2024) | 10.77 | 0.7324 | 17.45 | 0.7593 | 12.94 | 0.7186 | 13.60 | 0.7362 |
| 8 | Pika 1.0 (Pik, 2023) | 1.92 | 0.6856 | 12.77 | 0.7304 | 1.96 | 0.6410 | 5.33 | 0.6845 |
| 9 | Kling 1.0 (Kli, 2024) | 9.62 | 0.7308 | 19.15 | 0.7889 | 11.77 | 0.6869 | 13.33 | 0.7341 |
| 10 | Dream Machine (Lum, 2024) | 13.73 | 0.7646 | 19.15 | 0.8170 | 9.80 | 0.6772 | 14.09 | 0.7512 |
| 11 | Gen-3 Alpha (Gen, 2024) | 21.15 | 0.8102 | 17.02 | 0.7615 | 35.29 | 0.8261 | 24.67 | 0.8003 |

or may generate trivial motions. While I2V models generate more significant dynamics or transitions, they suffer from consistency and coherence issues, like the "rainbow" in Fig. 13. We also show additional qualitative comparisons of all the metrics considered in this work in Fig. 14-16. Existing metrics fail to address temporal compositionality and assign higher scores to static scenes without compositional changes.

# F    LIMITATION AND POTENTIAL SOCIAL IMPACTS

One limitation of our work is the discrepancy between our proposed metrics and human ratings. While TCR and TC-Score both demonstrate much higher ranking correlations with human judgments, there is still a need for having even more reliable and robust metrics for temporal compositionality. Our proposed evaluation metrics are not perfect. For example, VLMs still struggle with multi-image understanding. Besides, we rely on image-based assertion because strong video foundational models are lacking. To the best of our knowledge, temporal compositionality is still challenging in the context of video understanding. Therefore, future work could devise end-to-end video-based metrics when such a stronger VLM is available. In terms of potential social impacts, TC-Bench users and researchers should be aware of the potential abuse of text-to-video models. Hallucination issues and biases of generated videos should also be addressed. Future research should exercise caution when working with generated videos using TC-Bench prompts and ground truth videos.

Table 8: System prompt and first two in-context exemplars of the prompt.

**System Instruction:**
"Given a video description, generate assertion questions and paired frames to verify important components in the description. Each description describes a transformation/transition of an object's attribute, or an object's position or background. Identify the transition object, its start and end status/place, and other objects, and ask questions to verify them. Below are three examples showing three different types of transitions. Follow these examples and generate questions for the given descriptions."

**In-context exemplars 1:**
A chameleon changing from brown to bright green.
Transition object: chameleon, start: brown, end: bright green
other objects: None
- Check "Transition Completion"
Input: Frame 1
Q: Is there a brown chameleon?
Input: Frame 16
Q: Is there a bright green chameleon?
Input: Frame 9
Q: Is there a chameleon with its color in between brown and bright green?
Input: Frame 1, 5, 9, 13, 16
Q: Has the chameleon changed color from brown to bright green?
- Check "Transition object consistency"
Input: Frame 1, 6
Q: Aside from color difference, do Frame 1 and Frame 6 show the same chameleon?
Input: Frame 1, 11
Q: Aside from color difference, do Frame 1 and Frame 11 show the same chameleon?
- Check "Other objects"
None

**In-context exemplar 2:**
A man passing a ball from his left hand to his right hand.
Transition object: ball, start: left hand, end: right hand
other objects: man
- Check "Transition Completion"
Input: Frame 1
Q: Is there a ball on the man's left hand?
Input: Frame 16
Q: Is there a ball on the man's right hand?
Input: Frame 9
Q: Is the ball between the man's left hand and right hand?
Input: Frame 1, 5, 9, 13, 16
Q: Has the ball been passed from left hand to right hand?
- Check "Transition object consistency"
Input: Frame 1, 6
Q: Aside from position difference, do Frame 1 and Frame 6 show the same ball?
Input: Frame 1, 11
Q: Aside from position difference, do Frame 1 and Frame 11 show the same ball?
- Check "Other objects"
Input: Frame 1
Q: Is there a man with a ball in his hand in the image?
Input: Frame 1, 6, 11
Q: Do all the frames show the same man?

Table 9: The third in-context exemplar for assertion generation.

**In-context exemplars 3:** A bench by a lake from foggy morning to sunny afternoon.
Transition object: background, start: foggy morning, end: sunny afternoon
Other objects: bench, lake
- Check "Transition Completion"
Input: Frame 1
Q: Is the image showing a foggy morning?
Input: Frame 16
Q: Is the image showing a sunny afternoon?
Input: Frame 9
Q: Is the image showing a mix of foggy morning and sunny afternoon?
Input: Frame 1, 5, 9, 13, 16
Q: Has the background changed from foggy morning to sunny afternoon?
- Check "Transition object consistency"
None: background is an abstract concept without a physical form
- Check "Other objects"
Input: Frame 1
Q: Is there a bench by a lake in the image?
Input: Frame 1, 6, 11
Q: Do all the frames show the same bench and a lake?

*A chameleon turns from brown to green.*

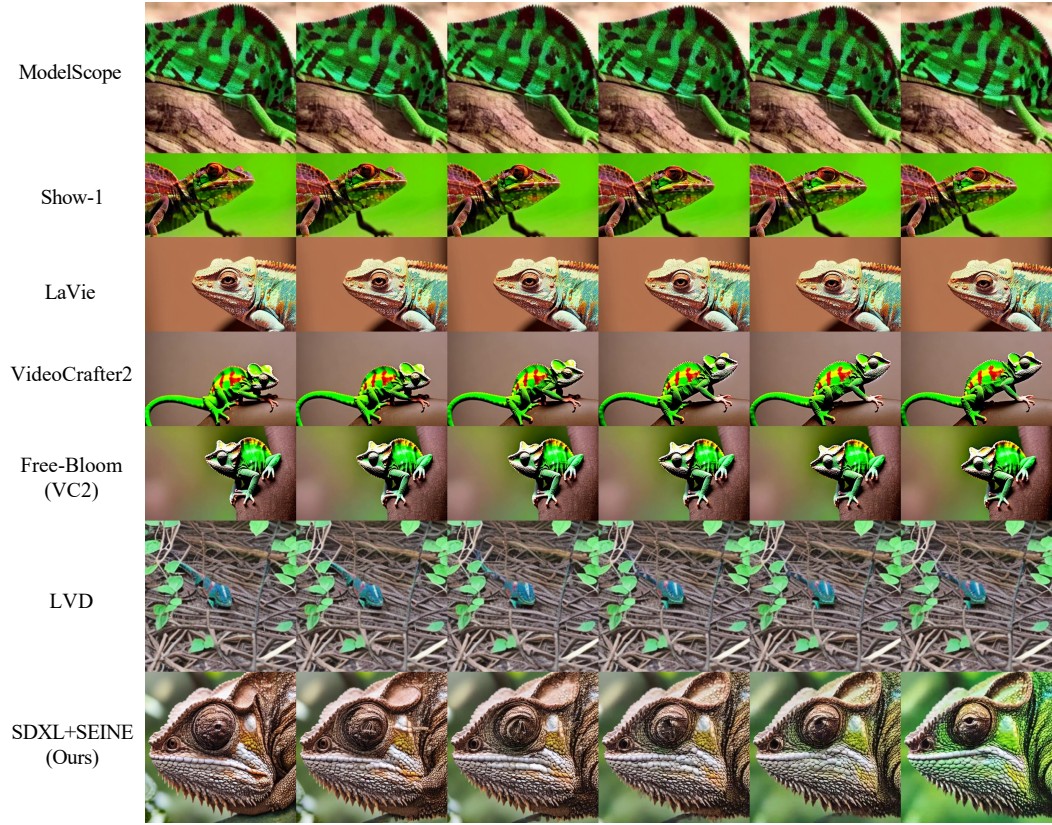

Figure 8: Additional qualitative examples of attribute transition of all T2V models on TC-Bench-T2V.

*A piece of fruit drops from a tree into a basket underneath.*

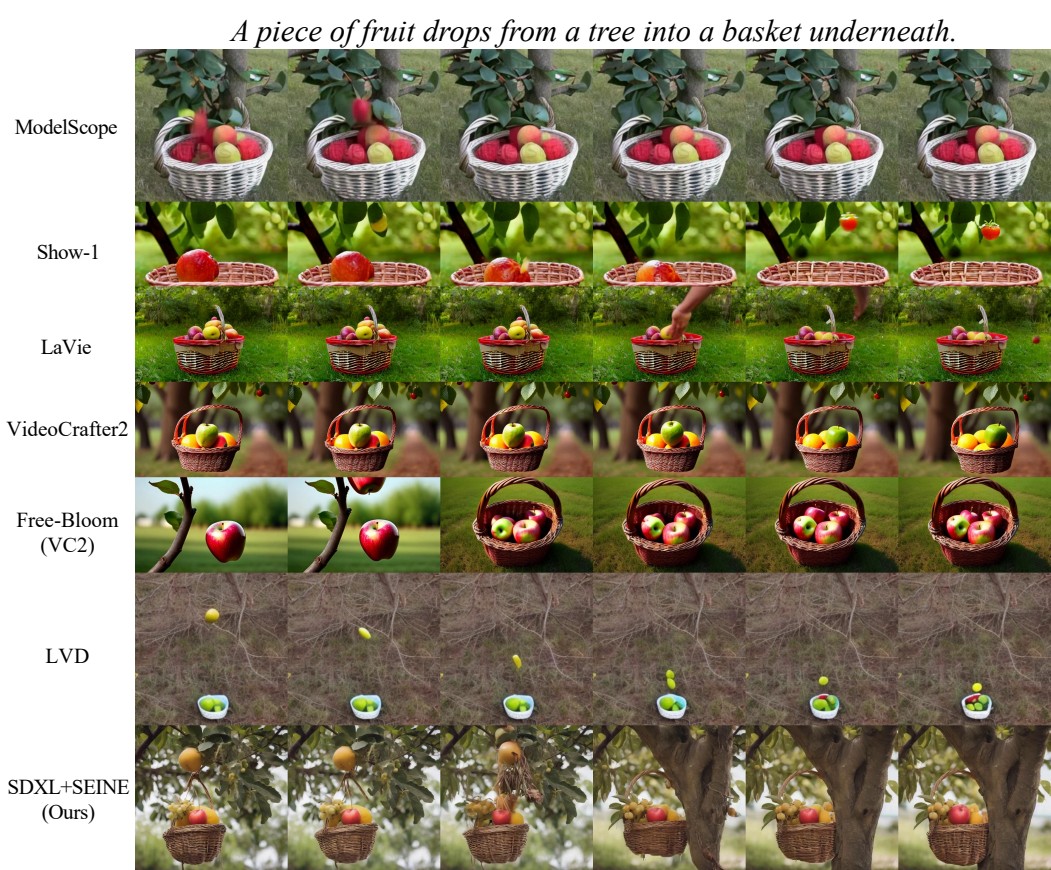

Figure 9: Additional qualitative examples of object relation change of all T2V models on TC-Bench-T2V.

*A forest changing from summer greenery to autumn foliage.*

Figure 10: Additional qualitative examples of background shifts of all T2V models on TC-Bench-T2V.

*A pink chameleon turns green.*

Figure 11: Additional qualitative comparison of attribute transition of direct T2V models and I2V models on TC-Bench-I2V.

*A bird holds a blue bottle cap and places it on the ground.*

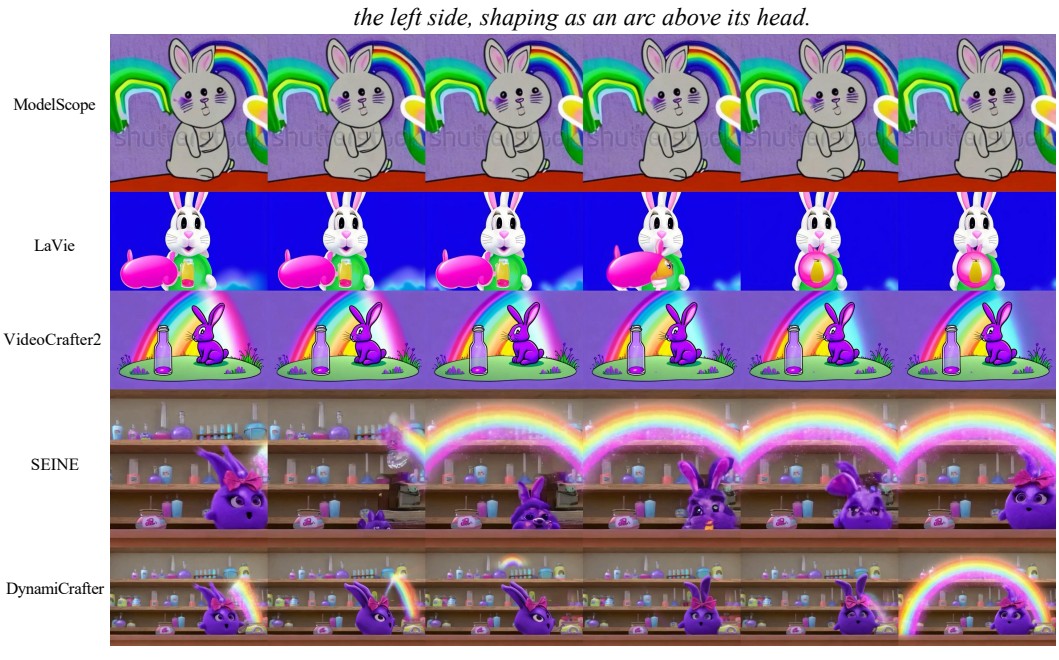

Figure 12: Additional qualitative comparison of object relation change of direct T2V models and I2V models on TC-Bench-I2V.

*A cartoon video of a purple bunny watching a rainbow streams from a bottle on the right to the left side, shaping as an arc above its head.*

Figure 13: Additional qualitative comparison of background shifts of direct T2V models and I2V models on TC-Bench-I2V.

|           | Top     | Bottom  |   |
|-----------|---------|---------|---|
| CLIP      | 0.2571  | 0.2496  | ✗ |
| ViCLIP    | 0.2970  | 0.2607  | ✗ |
| EvalCrafter | -0.2738 | -0.2740 | ✗ |
| UMT       | 4.0977  | 3.2676  | ✗ |
| TC        | No      | No      | ✓ |
| TC-Score  | 0.6777  | 0.8333  | ✓ |
| Human     | 2.333   | 3.0     |   |

An ice cream scoop melts from a round shape to a liquid puddle.

Figure 14: Additional qualitative comparison of different metrics on attribute transition.

|           | Top     | Bottom  |   |
|-----------|---------|---------|---|
| CLIP      | 0.2854  | 0.3293  | ✗ |
| ViCLIP    | 0.2726  | 0.2903  | ✗ |
| EvalCrafter | -0.2738 | -0.2721 | ✗ |
| UMT       | 4.0664  | 4.3867  | ✗ |
| TC        | No      | No      | ✓ |
| TC-Score  | 0.8333  | 0.5     | ✓ |
| Human     | 3.0     | 2.67    |   |

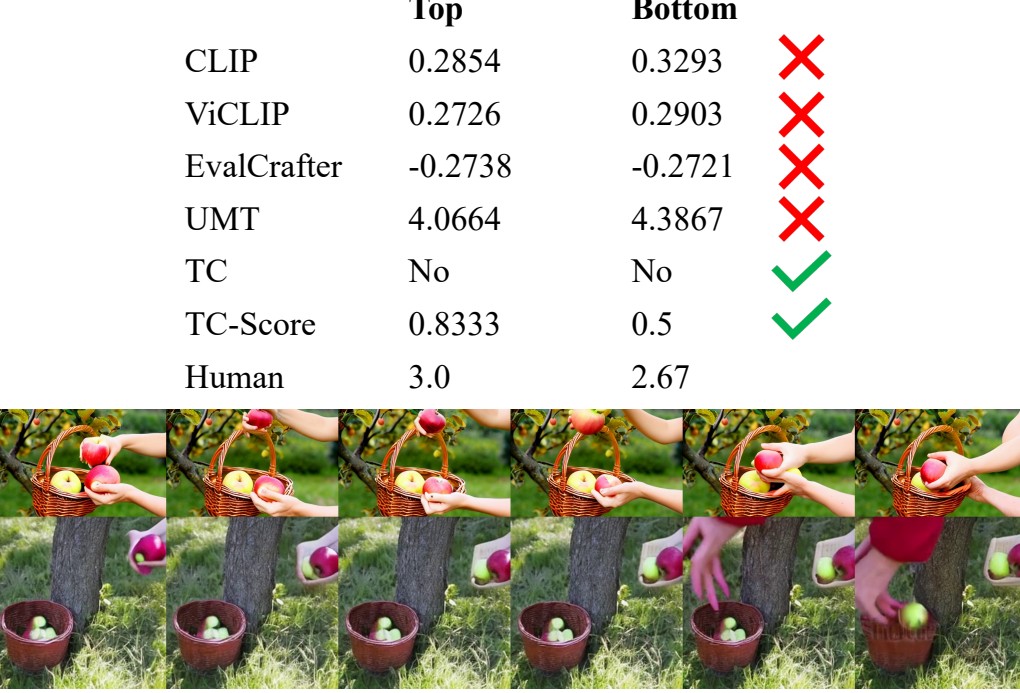

A woman picking an apple from a tree and placing it in a basket.

Figure 15: Additional qualitative comparison of different metrics on object relation change.

|  | **Top** | **Bottom** |  |
|---|---|---|---|
| CLIP | 0.2637 | 0.2200 | ✗ |
| ViCLIP | 0.2263 | 0.2168 | ✗ |
| EvalCrafter | -0.2800 | -0.2833 | ✗ |
| UMT | 4.1328 | 3.8301 | ✗ |
| TC | No | Yes | ✓ |
| TC-Score | 0.8333 | 1.0 | ✓ |
| Human | 2.0 | 5.0 | |

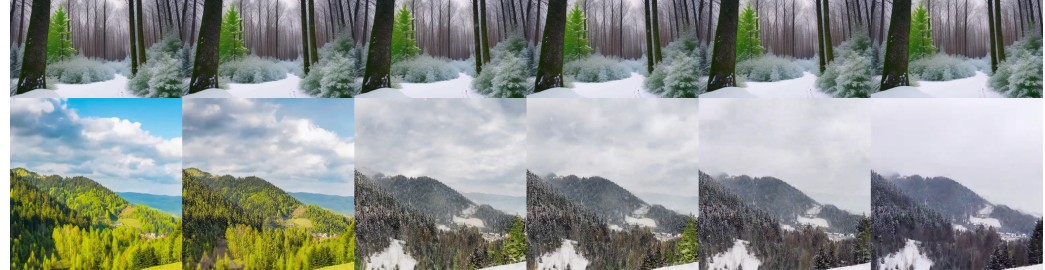

A forest changing from summer greenery to winter snow.

Figure 16: Additional qualitative comparison of different metrics on background shifts.

