# OpenReview forum: "TC-Bench: Benchmarking Temporal Compositionality in Conditional Video Generation"
_ICLR.cc/2025/Conference — Submitted to ICLR 2025_

### Official Review · Reviewer_zG8x · 2024-11-02

**Soundness:** 3
**Presentation:** 3
**Contribution:** 2
**Rating:** 3
**Confidence:** 4

**Summary:**

The paper presents a benchmark for evaluating video generation in temporal dimensions. The evaluation is composed of three dimensions: attribute transition, object relation change, and background shifts. VLMs (including GPT-4) are employed to get the quantitative results. The paper also contributes a simple baseline to improve the performance.

**Strengths:**

1. The paper is well-written and easy to follow.
2. The paper mitigates the shortcomings of existing video generation benchmarks, e.g., evaluating the generation quality of temporal dimension.
3. The paper conducts extensive evaluations of state-of-the-art video generation methods.
4. The authors implement a simple baseline to improve the quality.

**Weaknesses:**

1. The evaluation setting is still limited to short video generation. In long video generation, there are more than two attribute and relation changes. In the current setting, only binary state changes are considered, which hampers the extension to long video generation.
2. The evaluation relies heavily on the capability of VLMs. For example, the order of the results in Table 1 and Table 6 is inconsistent. If the method relies on GPT4, the cost of GPT4 hampers the wide usage of the proposed benchmark.

**Questions:**

Please see my questions in the weakness part.

---

### Official Review · Reviewer_ahB5 · 2024-11-03

**Soundness:** 3
**Presentation:** 3
**Contribution:** 3
**Rating:** 5
**Confidence:** 4

**Summary:**

This work introduces a benchmark for Temporal compositionality, addressing a previously overlooked and more challenging aspect of temporal compositionality in video generation. The proposed benchmark focuses on three key factors: attribute transitions, changes in object relationships, and background shifts. To evaluate these factors, the paper presents an evaluation metric and a generative method that demonstrates promising results.

**Strengths:**

This paper presents a highly specialized benchmark that addresses an editing problem currently not well-solved by existing text-to-video (T2V) and image-to-video (I2V) approaches. Its contribution is valuable, as it not only provides data but also includes evaluation metrics.

The writing of the paper is clear and well-structured, facilitating understanding of the proposed methods and results.

The evaluation metrics demonstrate alignment with human choices, reinforcing the relevance and applicability of the benchmark.

**Weaknesses:**

The Kling method could potentially be improved by incorporating first and last frames. I am curious whether this powerful commercial T2V/I2V model, when utilizing first and last frames alongside prompts, could resolve this issue, as it relates directly to the solution of the problem.

The length of the videos is relatively short, with most being under 5 seconds, which feels insufficient. Additionally, the number of videos is limited, with only 50 available for each scenario.

Figure 4 does not compare all methods. While it appears that Kling demonstrates strong instruction-following capabilities regarding object relationship changes, this is not reflected in the numerical values in Table 1. Are there any examples of bad cases available? The Gen3 model seems to perform well with background shifts. It appears that attribute transitions are the most challenging aspect. Thus, would incorporating first and last frames help mitigate this issue?

**Questions:**

Does the benchmark account for scenarios where multiple factors change simultaneously? For instance, transitioning from a yellow person sitting in a green car by the seaside to a yellow dog sitting in a yellow car in the desert. Can the evaluation method address such cases?

In the context of attribute transitions, could variations in materials and textures lead to changes in object ID evaluation?

Might changes in video framing also affect the accuracy of the VLM (Vision-Language Model) or evaluation metrics?

The supplementary materials lack a README file, making it unclear how to interpret the evaluation results of "eval_results" folder.

---

### Official Review · Reviewer_p1Js · 2024-11-04

**Soundness:** 3
**Presentation:** 3
**Contribution:** 2
**Rating:** 5
**Confidence:** 4

**Summary:**

This paper proposes TC-Bench, a new benchmark designed to assess the Temporal Compositionality of video generation models. TC-Bench is divided into two components: TC-Bench-T2V, which includes 150 prompts for evaluating Text-to-Video (T2V) models across a spectrum of attributes, actions, and objects, defining the initial and final states of scenes; and TC-Bench-I2V, comprising 120 prompt-video pairs that serve as ground truth videos and reference data for Image-to-Video (I2V) models. The metrics introduced in this study demonstrate a significant correlation with human judgments, enhancing the evaluation of temporal compositionality.

**Strengths:**

1. The focus on temporal compositionality in video generation is both novel and important, given the rapid advancements in conditional video generation. The benchmarks and quantitative experiments presented are valuable additions to the field.

2. The methodology is robust, featuring comprehensive experiments with detailed explanations, facilitating replication and further study.

3. The paper is well-written, structured for clarity.

**Weaknesses:**

1. The sizes of the test sets for both T2V (150) and I2V (120) benchmarks are relatively small, which may limit their ability to comprehensively analyze the capability for temporal compositionality.

2. There is a lack of analysis on the distribution of the test sets, raising concerns about their representativeness of real-world scenarios relevant to the tasks.

3. The evaluation is limited to only two methods, SEINE and DynamiCrafter, within the I2V category. This might not provide a full perspective on the field, given the variety of available I2V methods.

4. The influence of the structure and length of the input prompts on video generation quality is a critical aspect that remains unexamined, which could impact the effectiveness of the benchmarks.

**Questions:**

n/a

---

### Official Review · Reviewer_dEqD · 2024-11-04

**Soundness:** 2
**Presentation:** 2
**Contribution:** 3
**Rating:** 6
**Confidence:** 4

**Summary:**

The paper focuses on assessing the temporal compositionality of video generation models, featuring carefully designed text prompts and ground truth videos, along with two robust evaluation metrics: TCR and TC-Scores. The benchmark is important as it better reflects the temporal dynamic performance of videos compared to previous benchmarks for video generation. The article presents three scenarios for temporal compositionality and conducts extensive baseline methods, ranging from direct T2V models to I2V models. This benchmark provides new perspectives for evaluating and improving video generation tasks.

**Strengths:**

The temporal compositionality proposed in the article is a significant evaluation aspect of video generation that  was not addressed in previous benchmark papers. The proposed benchmark can serve as a supplement to current video generation evaluations, promoting advancements in the field of video generation.

**Weaknesses:**

1. Regarding the evaluation of temporal compositionality, the article proposes three scenarios. Whether these scenarios are comprehensive and cover all possibilities requires further discussion and analysis.
2. The article lacks more detailed descriptions regarding prompt design, such as topics and length distribution.

**Questions:**

1. How many prompts and ground truth videos are there in total, specifically for T2V and I2V? Please provide a chart to present this more clearly.
2. How is the topic distribution of the prompts considered? It is recommended to provide a specific diagram categorizing the topic types.

---

### Meta-Review · Area_Chair_zXS5 · 2024-12-16

**Metareview:**

During the review procedure, reviewer pointed out that the authors insert links that can reveal their names in the appendix (Line 824, 825).
 This paper should be rejected without further discussions.  The AC recommend rejection.

**Additional Comments On Reviewer Discussion:**

As this paper violates Anonymity guidelines，it should be rejected without further discussions.

---

### Decision · Program_Chairs · 2025-01-22

Reject